# Structure Determination by Single-Particle Cryo-Electron Microscopy: Only the Sky (and Intrinsic Disorder) is the Limit

**DOI:** 10.3390/ijms20174186

**Published:** 2019-08-27

**Authors:** Emeka Nwanochie, Vladimir N. Uversky

**Affiliations:** 1Department of Molecular Medicine, Morsani College of Medicine, University of South Florida, Tampa, FL 33612, USA; 2USF Health Byrd Alzheimer’s Research Institute, Morsani College of Medicine, University of South Florida, Tampa, FL 33612, USA; 3Laboratory of New Methods in Biology, Institute for Biological Instrumentation, Russian Academy of Sciences, Pushchino 142290, Moscow Region, Russia

**Keywords:** structural biology, protein structure, 3D-structure, structural analysis, X-ray crystallography, NMR spectroscopy, Cryo-electron microscopy, single-particle Cryo-EM, Intrinsically disordered protein, intrinsically disordered region

## Abstract

Traditionally, X-ray crystallography and NMR spectroscopy represent major workhorses of structural biologists, with the lion share of protein structures reported in protein data bank (PDB) being generated by these powerful techniques. Despite their wide utilization in protein structure determination, these two techniques have logical limitations, with X-ray crystallography being unsuitable for the analysis of highly dynamic structures and with NMR spectroscopy being restricted to the analysis of relatively small proteins. In recent years, we have witnessed an explosive development of the techniques based on Cryo-electron microscopy (Cryo-EM) for structural characterization of biological molecules. In fact, single-particle Cryo-EM is a special niche as it is a technique of choice for the structural analysis of large, structurally heterogeneous, and dynamic complexes. Here, sub-nanometer atomic resolution can be achieved (i.e., resolution below 10 Å) via single-particle imaging of non-crystalline specimens, with accurate 3D reconstruction being generated based on the computational averaging of multiple 2D projection images of the same particle that was frozen rapidly in solution. We provide here a brief overview of single-particle Cryo-EM and show how Cryo-EM has revolutionized structural investigations of membrane proteins. We also show that the presence of intrinsically disordered or flexible regions in a target protein represents one of the major limitations of this promising technique.

## 1. Introduction: Brief Historic Overview of Methods of Structural Biology

When faced with a problem, a reduction to its component parts is generally a common practice in science. Oftentimes, fundamental insights are readily obtained by piecing the individual parts back together. A deep understanding of the overall structure of a problem informs potential strategies that could be undertaken in solving the underlying constraint. This rationale particularly applies to the growing field of structural biology, recognized approximately 70 years ago following structural elucidation of the atomic structure of DNA and other globular proteins [1]. Built on the collective efforts of researchers in this field, it is now possible to determine the structural architecture of biologically relevant macromolecules and assemblies (proteins and nucleic acids) that play very important roles in the overall coordination of human physiology. It is important to note that the relationship between protein structure and function is not always one of complete linearity, and this complexity of the structure–function relationship is commonly exemplified by the class of intrinsically disordered proteins and polypeptides [2,3,4,5,6,7,8]. Hence, structural information alone is not sufficient to make conclusive remarks on protein functionality. There is usually a trade-off between structure and function.

Watson and Crick’s characterization of the tertiary structure of DNA presented a huge milestone in 1953. This was only possible through X-ray crystallographic data sourced from Rosalind Franklin and Maurice Wilkins [9]. In this respect, X-ray crystallography became the primary tool of choice for the determination of high-resolution atomic models of biological complexes for many decades. A macromolecule of interest is purified and crystallized in a pre-determined crystallization solution specific to the target under study. The crystallized particle is shot with X-ray beam at different angles, producing a unique 2D diffraction image, immediately visualized on a detector. The X-ray irradiation is scattered by atoms of the particle under investigation, and characteristic spots in the diffraction map represent the results of these X-ray-atom interactions.

Subsequently, a mathematical Fourier transformation is applied, which converts the resulting diffraction pattern into a 3D electron density map. Although X-ray crystallography is an extensively utilized technique in structural biology, it is not without limitations. Sample heterogeneity and biomolecular flexibility remains a great challenge in crystal formation, hence predisposing the need for the development of novel techniques and multi-disciplinary approaches that are able to surmount existing limitations of the technique. Variations in the conception of change do exist amongst individuals, however it does not take away the fact that change is an extremely dynamic phenomenon and, quite frankly, inevitable. Within the scientific community, change remains the trigger for innovation and more often than not, new technologies and systemic improvements are frequently being researched. The abundance of structural information gathered over the years from X-ray crystallography has provided a wealth of knowledge, which has laid the foundation for advancing the development of other structure determining biophysical tools. Following World War II, Felix Bloch and Edward Purcell received a joint Nobel Prize for making independent discoveries in nuclear magnetic resonance (NMR) spectroscopy. The establishment of the technique as an investigative tool applicable across disciplines took approximately half a century [10]. Both X-ray crystallography and NMR spectroscopy were adopted over time as primary methods for structure determination at atomic resolution. The many limitations posed by the former were significantly surmounted by the latter. Not only does solution NMR give 3D structural information of bio-macromolecules, it also presented an unprecedented approach to probe dynamic molecular interactions (e.g., protein–protein, protein–ligand, or protein–nucleic acid interactions) and characterize conformational ensembles at physiological conditions. NMR finds its most successful application in the study of the dynamics of intrinsically disordered proteins (IDPs) [11,12,13,14,15,16,17,18,19,20,21,22,23,24,25,26]. These proteins and protein regions exist commonly in eukaryotes. There is an increasing need for their studies, especially because they play a major role in molecular recognition and cell signaling [27]. NMR experiments are able to yield accurate and sensitive information of the interaction affinity between species and the site of interaction between a protein and its binding partner [11,12,13,14,15,16,17,18,19,20,21,22,23,24,25,26]. For a protein NMR experiment, purified protein is strategically placed in a strong magnetic field. The resultant unique set of resonances generated are then analyzed in order to create a list of atomic nuclei related to each other in terms of proximity and stereochemistry. Finally, a 3D structural model of the protein is built from data generated which reveals the individual location of the atom. Large molecular weight proteins are often difficult to characterize by NMR spectroscopy due to their tendency to display overlapping peaks in the NMR spectra, making data interpretation a daunting task. Therefore, for a long time, this technique was limited to small or medium size peptides and proteins. Large size target proteins pose a logical hurdle for their structural assignment by NMR (traditionally, the molecular mass of most proteins of which their structures were solved by NMR was <25 kDa [28]). There are, however, recent advances in this field that are pushing the upper molecular mass boundary. These advancements has enabled structural characterization of large proteinaceous machines with molecular masses of hundreds of kilodaltons [29], e.g., 20S proteasome [30]. Among the notable developments is the utilization of a transverse relaxation optimized spectroscopy (TROSY). This is a technique for suppression of transverse relaxation in multidimensional NMR experiments [31] combined with specific labeling schemes [32]; the use of selective protonation of methyl groups in highly deuterated background [33], incorporation of (1H-δ methyl)-leucine and (1H- γ methyl)-valine into 15N-, 13C-, 2H-labeled proteins [34], or using reductive 13C-methylation of lysines [35]. Additionally, NMR studies have to be conducted on isotope-labeled proteins; a simple process for heterologous protein expressions in *Escherichia coli* or in cell free expression systems. However, isotopic labeling for proteins that cannot be produced in these systems proves very challenging [36]. 

Cryo-Electron Microscopy (Cryo-EM) is a new and powerful technique for the elucidation of the 3D structure of biomolecules and biological assemblies, through single-particle imaging of non-crystalline specimens. In as much as Cryo-EM was developed around the 70s, the scientific community only recently started to appreciate the technique. Due to significant improvements in the technique, sub-nanometer atomic resolutions (<4 Å) became a reality [37].

In 2017, Jacques Dubochet, Joachim Frank, and Richard Henderson were awarded a Nobel Prize for their contributions to the development of Cryo-electron microscopy (Cryo-EM). In order to acquire structural details of a particle of interest, electrons are shot at the particle frozen in solution. Structural biologists are quickly adopting cryo-EM because it proffers a way to image large molecular weight biomolecules with flexible structure, which cannot be easily deduced form protein crystals [38].

Development and subsequent advancement of various protein imaging tools enrich the protein data bank (PDB), a large repository for information on protein tertiary structures. Since its inception, X-ray crystallography is believed to have impacted the characterization of >112,000 protein structures housed within the PDB. For this reason, this approach is ranked as the most commonly used technique for protein structure elucidation. NMR spectroscopy takes second place (approximately 10,500 structures) in adding to the continually expanding PDB. At the same time also, electron microscopy contributed ~1200 structures. Although, the last 5 years witnessed a major leap in the number of yearly deposited PDB structures resolved by cryo-EM [39].

## 2. Cryo-EM: Single-Particle Analysis (SPA)

The impact of cryo-EM through improved sample preparation methods [40], application of direct electron detector (DED), and advanced computational algorithms with the capacity to accurately resolve images of structurally heterogeneous specimens initiated the so called “resolution revolution” era [41,42,43]. A general schematic of cryo-EM shown in Figure 1 involves specimen preparation, low dose image collection, and model building. These steps are common to all types of cryo-EM experiments. Single particle analysis (SPA) and Sub-tomogram averaging (STA) comprise two models for data collection and 3D reconstruction [44].

Single-particle analysis (SPA) computationally averages multiple 2D projection images of the same particle for 3D reconstruction [44]. For this reason, large numbers of single particles are collected to compensate for poor signal-to-noise (SNR) correlated with low dose imaging of cryo-EM [1]. The collection of images at high electron doses typically subjects the specimen to radiation damage, whereas imaging at low electron doses introduces some irregularities, which account for the major part of noise production. The alignment and averaging of many 2D micrographs over time will drastically reduce background noise. Progress associated with Cryo-EM 3D-reconstruction would not have been possible without recent advances in computational power [1]. Alternatively, sub-tomogram averaging (STA), unlike SPA, performs tomographic reconstruction of 3D structures embedded in amorphous ice [44].

### 2.1. Specimen Preparation for Single Particle Cryo-EM Analysis

An important factor relating to any experimental study is the consideration of the quality of each sample to be analyzed. Accordingly, the quality of experimentally generated data is directly proportional to the quality of the sample being analyzed. Cryo-EM samples are prepared to fully maximize image quality. Particles displaying complex structural heterogeneity should be reduced to a more homogenous subset prior to cryo-EM analysis. Conformational heterogeneity represents a huge challenge in cryo-EM, particularly because molecules are not tightly organized into crystals, as is the case with X-ray crystallography. Reduction of conformational variability in a ribosomal particle from >1.6 million particles to a homogenous subset of ~400,000 yielded a resolution of <2.9 Å [1]. The use of conventional gel-filtration chromatography alone is considered to be insufficient in making a particular sample amenable to cryo-EM studies. An intact complex could potentially contain compositionally heterogenous sub-complexes. It is also believed that a homogenous complex can occur in several different conformations [37]. 

Negative staining technique serves as a quick optimization step in preparing samples for cryo-EM analysis [37]. The sample is blotted on to an EM grid surface, enclosed within a medium or high electron density. High contrast is observed due to existing relative electron density between the sample and the surrounding stain. The less dense particle scatters electrons at a low pace than the electron-rich staining medium, thus causing the latter to appear darker. Despite being limited by resolution, negative staining method of sample preparation generally improves the quality of the sample to be studied by electron microscopy [45]. It diminishes conformational heterogeneity by stabilizing preferred conformations of the sample. This improves image contrast and provides a quick way to evaluate sample purity [37,45].

Chemical crosslinking with glutaraldehyde or an on-column crosslinking approach have proven to be quite efficient in the management of compositional heterogeneity of samples [46,47]. It is worth noting that any one of the crosslinking techniques generally is likely to introduce unwanted features into the sample mix. For instance, the most compact configuration of a protein ensemble is most likely to be favored over the remaining conformational types. More so, flexible terminals or loops within a complex can engage in interactions resulting in the formation of a non-canonical structure [48]. Despite these drawbacks, crosslinking is rather widely used in cryo-EM as a powerful tool for the reduction of sample heterogeneity [49,50,51,52,53,54,55,56]. 

Conformational heterogeneity is a lot more difficult to handle, especially when a protein contains a few flexible domains. In this case, structure determination may be limited to negative staining EM. Optionally, chemical crosslinking may be applied to lessen the burden of conformational heterogeneity, albeit the biological relevance of structural data obtained will require full authentication. Furthermore, conformational heterogeneity can be tackled by deliberately holding the protein of interest in a unique functional state. This can be achieved by inducing certain interactions between the target protein and compounds capable of altering the functionality of the protein under investigation. Such compounds include inhibitors, ligands, and co-factors [57,58,59,60,61,62,63,64,65].

Recent advances in the detector technology, coupled with the development of sophisticated algorithms for image processing have greatly impacted the efficiency of managing structural heterogeneity. Some of the biochemical manipulations previously discussed should not be completely discarded, because an initial reduction of structural heterogeneity in a sample prompts a less tedious downstream image processing. The vital role played by negative staining in sample optimization cannot be emphasized enough. Nonetheless, it does present a likelihood of introducing false heterogeneity. For this reason, sample vitrification was developed as an alternative method of sample preparation to eliminate possible issues of heterogeneity [37]. Dubochet et al. developed cryo-EM in an attempt to prevent sample dehydration when placed in the vacuum of an electron microscope. Flash-freezing a biomolecule preserves its native architecture [38]. An implication of this process is a high-resolution image produced from sustained relative stability of molecules making up the specimen as they interact with the electron beam [38].

The specimen for cryo-EM analysis is frozen rapidly in solution, before being placed in the column of the electron microscope. The column is maintained at a high vacuum/low temperature condition in order to preserve the amorphous consistency of the frozen sample and to mitigate the effect of radiation damage. The restriction in movement of molecules making up the vitrified sample, caused by the radiation-induced “cage effect”, ultimately prevents molecular collapse [66]. Ice adopts various geometric modalities relative to temperature variations. For example, cubic ice readily forms at temperatures between −123 °C to −148 °C, whereas temperatures over −103 °C induce the formation of hexagonally-shaped ice crystals [67]. However, instantly freezing a specimen limits the potential to assume one of the possible crystalline transition states and amorphous ice is formed instead [66,68]. Importantly, the sample must be maintained at extremely low temperatures that discourage the formation of crystalline ice. Cryo-EM analysis of tobacco mosaic virus presents a particularly unique case. A high resolution image of the virus was observed in the crystalline form compared to that fixed in amorphous ice. So far in cryo-EM, various cryogenic substances, such as liquid helium, liquid ethane, and liquid nitrogen, have been used at some point. However, liquid nitrogen is commonly being used for cryo-EM imaging because of the uniformity in the data quality generated as a result [66]. 

### 2.2. Data Acquisition and Image Processing

A lot of time is generally spent in processing of large amounts of 2D micrographs, which constitute the main workload in any single particle cryo-EM project [69]. Repetitive movement of the mechanical stage during data acquisition amounts to a significant time lag. Data acquisition efficiency diminishes over time due to the accumulation of process waste. One way of overcoming this time constraint is by adjusting the electron beam trajectory to gain control of the data acquisition area [70]. Doing this may significantly reduce lagging time, but at the same time introduces “comatic aberrations”, which affects resolution. In this respect, image quality is achieved through the use of image processing software that account for aberrations and beam-tilt [71,72,73,74,75]. With improved data collection strategies, analysis of complex samples is possible. The images of particles exhibiting high conformational variability can be collected while they are being maintained at a stable tilt position. Of course, this method is experimentally complicated and potentially produces a noisy image due to increased sample thickness [76,77].

Image collection using phase plates is gaining popularity fast in the field of cryo-EM due to their ability to improve image contrast. Volta Phase Plate (VPP), a type of phase plate technology, yielded the first high resolution cryo-EM 3D structures of hemoglobin [78], G-protein coupled receptor [79], streptavidin [80], and the nucleosome [81]. VPP uses a material film in its beam track that is believed to produce poor resolution images. Therefore, new phase plates, based on high-intensity laser technology, are currently under development. This new technology is likely to replace VPP as it proposes no loss in image information and a constant phase shift [82].

It is often difficult to obtain good quality images from vitrified biological specimen, despite the relatively high resolution (~2 Å) afforded by a transmission electron microscope (TEM). Poor resolution images result from low electron dose sampling. Therefore, an optimal adjustment of the cryo-EM operating conditions is required to increase phase contrast [37]. Automated particle picking programs, such as Relion-autopick [73,83], DeepPicker [84], AutoPicker with ViCer (View Classifier) [84], AutoCryoPicker [85], SPHIRE-crYOLO [86], and several others, are presently available. Algorithms based on maximum likelihood consistently have improved image quality, thereby dominating the field of cryo-EM [87,88]. Reports have shown that these extremely user-friendly computer programs remain the source of attraction for interested users as these programs are packaged to incorporate a comprehensive image processing workflow [66].

### 2.3. 3D-Reconstruction and Structure Validation

Single particle 3D reconstruction is almost similar to assembling the pieces of a puzzle. Multiple 2D micrographs from a single particle are considered the puzzle pieces. These pieces need to be rationally organized into a representative 3D model, hence solving the puzzle. The “projection-slice theorem” generally forms the basis of image transformation [89]. It states that a 2D Fourier transform is equivalent to the central slice cutting through the origin of the reference 3D Fourier transform, and that the direction of the 2D projection lies at a right angle to the 3D slice [89]. Consequently, having prior knowledge of the correct orientation of multiple 2D projections of a single particle creates a possibility for predicting its 3D model [89].

When performing 3D reconstruction for single particle cryo-EM, one must consider two important factors: 

1. Imaging at low electron dosage (20–40 electrons/Å2) prevents radiation damage, but severely lowers SNR; 

2. The relative position of images of a single particle are unknown. 

Hence, a “projection matching” tool that compares experimentally determined images (initial reference) against a computationally derived 3D reference structure (reprojection images) is applied [76,89,90]. Supposing all possible experimental spaces in terms of slice directionality are potentially being covered, projection matching-based 3D reconstructions will produce a better structure than its experimental counterpart [89,90]. A reconstructed 3D model of high resolution, which reflects a true structure of the particle under study, should be expected through repetitive projection matching [89,90,91,92].

The degree of similarity between an initial reference and the particle being studied is greatly emphasized in 3D reconstruction [66,93,94]. This is because available projection-matching tools are not optimized for global analysis and are limited to short coverage distances [66,90]. Hence, if the initial reference image does not match the structure of the particle under investigation, the program is most likely to sample incorrect data points. Some upgrades in the projection-matching algorithms based upon random sampling have been applied as well, although accuracy is still not guaranteed [89,90,92,95]. Available crystallographic data of particles of interest (not used as an initial reference) can be used to validate 3D reconstructed models from single particle analysis. A major validation issue erupts if there are no available reference structures and if the reconstruction model is of very poor resolution. It naturally becomes difficult to identify truly representative 3D reconstruction data. For this reason, a powerful validation approach that looks into prior information of image pairs recorded at varying angular tilts is required [66].

## 3. Single-Particle Cryo-EM Analysis of Membrane Proteins

The discussion below presents some evidence on how single-particle cryo-EM has impacted the collection of structural information pertaining to several membrane proteins. Here, some major milestones in structural characterization of large transmembrane proteins by single-particle Cryo-EM are outlined.

### 3.1. Ryanodine Receptor (RyR) Channels 

RyR is an integral membrane scaffolding protein that resides in the sarcoplasmic/endoplasmic reticulum (SR/ER) membrane of muscle cells, serving as controllers and regulators of the excitation-contraction coupling of skeletal and cardiac muscles [96,97,98,99,100]. RyR channels are the largest homo-tetrameric ligand-gated Ca^2+^ receptors with a size of >2.2 MDa, which function in calcium-dependent signaling of muscle contraction, being responsible for the rapid release of Ca^2+^ ions from the SR/ER to the cytoplasm [96,97,98,99,100]. In mammals, there are three isoforms of RyR with a sequence identity of ~70% [101], with RyR1 and RyR2 being preferentially expressed and functional in skeletal and cardiac muscles, respectively, and with distribution and functionality of RyR3 being poorly characterized (although originally, it was found in the brain) [102,103]. In humans, RyR1 (UniProt ID: P21817), RyR2 (UniProt ID: Q92736), and RyR3 (UniProt ID: Q15413) have sequences of 5038, 4967, and 4870 residues, respectively. Therefore, the assembled homo-tetrameric RyRs are gigantic, being characterized by a mushroom-shaped architecture, with a very large cytoplasmic domain (>4500 residues) and a ~500-residue-long C-terminally located trans-membrane channel domain. The cytoplasmic region of each protomer of the RyR channel includes nine separate domains, the N-terminal domain (residues 1-631 in rabbit RyR1, UniProt ID: P11716), three SPRY domains (residues 582–798, 1014–1209, and 1358–1571, respectively), P1 (residues 800–1000) and P2 (residues 2734–2940) domains, Handle (residues 1651–2145), Helical (residues 2146–2712 for HD1 and 3016–3572 for HD2), and Central (residues 3668–4251) domains [101].

The RyR channels are undeniably critical for survival, thereby motivating their in-depth structural and functional studies. High-resolution 3D structures of these channels are expected to provide a good starting point for understanding the molecular mechanism surrounding RyR channel gating in both normal and abnormal states [96,97,98,99,100]. Large size and conformational variability of the RyR channels make them unamenable to X-ray and NMR spectroscopy. Thin-section electron microscopy was the first method used to have a glance at the structural organization of the RyR channel. This analysis produced a poor resolution image, showing a structure resembling a large foot [96,97,98,99,100]. Negative stain Cryo-EM was first used to model the 3D structure of the RyR channel at ~ 40 Å resolution, giving insight into its mushroom-shaped morphology [104]. 

The incredibly large size (2.3 MDa), coupled with the ability to be purified relatively easily put RyR amongst the first non-icosahedral proteins to be structurally characterized by single-particle Cryo-EM. Structural elucidation of the RyR channel progressed alongside advances in the single-particle EM. By capturing the molecule in a more natural hydrated state, this permitted the reconstruction of the first 3D density map of RyR at resolutions between 20 and 30 Å. Advanced detector technology and 3D classifications encouraged further resolution refinement of three more RyR structures of up to 3.8 Å [36,101,105]. 

Figure 2A–C represent the overall shape of rabbit RyR1 in three projections in the space-fill mode, whereas Figure 2D–F show the same projection in the cartoon mode. These images were generated using single-particle Cryo-EM analysis of rabbit RyR1 homo-tetramer complexed with a human chaperone, peptidyl-prolyl cis-trans isomerase FKBP1B (PDB ID: 5T15, [105]) and clearly demonstrate the complex structural organization of this important channel. Based on the single-particle Cryo-EM analyses, about 70% of 2.2 MDa of this protein were structurally resolved [101]. Among regions that escaped structural assignment in single-particle Cryo-EM experiments are residues 1–11; 86–96; 126–129; 226–228; 324–327; 361–370; 427–435; 500–505; 652–659; 795–798; 947–959; 1170–1178; 1262–1271; **1298**–**1431**; 1442–1446; 1468–1471; 1522–1524; 1561–1576; 1751–1760; 1785–1798; **1869**–**1924**; 2073–2092; 2169–2171; 2217–2225; 2308–2324; 2363–2374; **2387**–**2418**; 2438–2448; 2511–2513; 2538–2543; 2831–2848; 3681–3693; 3734–3749; 3853–3876; 4065–4070; 4131–4134; **4252**–**4544**; 4584–4637; 4743–4764; and 5034–5037 [101]. Although the majority of these regions are relatively short, some of them (shown in bold font) are rather long, with lengths exceeding 30 residues. Curiously, Figure 2G shows that all long regions with unresolved structures and a large part of short such regions coincide with regions predicted to be intrinsically disordered (i.e., regions with predicted intrinsic disorder scores exceeding the 0.5 threshold) by several commonly used disorder predictors. The majority of the remaining unresolved regions overlapped or were located in the close proximity of flexible regions (i.e., regions with predicted disorder scores ranging from 0.2 to 0.5). These are important observations indicating that similar to X-ray crystallography, the presence of disordered or highly flexible regions represents a bottleneck for structural characterization of proteins by single-particle Cryo-EM. 

### 3.2. ATPases/V-ATPases 

ATP synthases (ATPases) are a class of intrinsic membrane bound proton pumps that couple energy derived from hydrolyzing ATP to push protons uphill against a concentration gradient across the plasma membrane. There are three major classes of ATPases [113], such as archaeal A-Type ATP synthase [114,115], F-Type ATPase found in bacterial plasma membranes, mitochondrial inner membranes, and in chloroplast thylakoid membranes [116], and vacuolar-type H^+^-ATPase (V-ATPase) found in plasma membrane of eukaryotic specialized cells and within the membranes of organelles, such as lysosomes, endosomes, and secretory vesicles [117,118], and in some bacteria [119]. Eukaryotic vacuolar-type ATPases (V-ATPase) are the most complex (900 KDa) and the most evolved members of this protein family. The eukaryotic V-ATPase includes 14 subunits (A_3_B_3_CDE_3_FG_3_Hac_x_c′_y_c″_z_de), which form the cytosolic V_1_ (which includes subunits A-H and is responsible for ATP hydrolysis) and the integral V_0_ domains (which contains subunits a, d, e, c, and c″ and carries out proton transport) [120]. In the eubacterium *Thermus thermophiles*, the V-ATPase is composed of nine different subunits with a stoichiometry of A_3_B_3_CDE_2_FG_2_IL_12_ [121]. 

ATPases/V-ATPases function in various biological processes, such as membrane trafficking, protein processing and degradation, ATP-coupled transportation of small compounds, bone resorption, and urinary acidification [117,122]. V-ATPases have been reported to be associated with some disease progressions of the kidney, viral infections, osteoporosis, neurodegeneration and cancer, thus making it a potential molecular target in cancer drug discovery, justifying a need for good structural insight [117,122]. A unique characteristic of V-ATPases is the reversible association/disassociation of the V1 and V2 regions. This does not require new protein expression to occur. 

Insufficient sample, very complex structure, as well as the propensity for V-ATPase disassociation into its component subunits have greatly interfered with crystallizing the intact enzyme. A hybrid method has been applied to get structural insights. For example, small angle X-ray scattering and NMR yielded atomic models of component subunits, while single particle Cryo-EM yielded density maps of the intact enzyme [122]. 

V-ATPase from a thermophilic bacterium was the first membrane protein structure to be characterized by single-particle cryo-EM at sub-nanometer resolution (<8.5 Å) [123]. The corresponding 3D structure of this protein machine is shown in Figure 3A (PDB ID: 5GAR, [123]). Recent advances in cryo-EM techniques triggered the determination of high-resolution 3D density maps of three V-ATPases structures from *Saccharomyces cerevisiae* at resolutions between 6.9 Å and 8.3 Å [124]. These sub-nanometer resolution images (e.g., see Figure 3B showing a 6.9 Å resolution structure of this protein, PDB ID: 3J9T, [124]) gave solid insights into the different rotational states of V-ATPase, including its functional dynamics during each enzymatic cycle [36]. Similar to the situation described for the RyR proteins, although single particle Cryo-EM yielded 3D structure of intact eukaryotic V-ATPase comprising V-type proton ATPase subunit D (UniProt ID: P32610, chain M), V-type proton ATPase subunit F (UniProt ID: P39111, chain N), V-type proton ATPase catalytic subunit A (UniProt ID: P17255, chains A, C, and E), V-type proton ATPase subunit B (UniProt ID: P16140, chains B, D, and F), V-type proton ATPase subunit d (UniProt ID: P32366, chain Q), V-type proton ATPase subunit G (UniProt ID: P48836, chains L, H, and J), V-type proton ATPase subunit E (UniProt ID: P22203, chains K, G, and I), V-type proton ATPase subunit H (UniProt ID: P41807, chain P), V-type proton ATPase subunit a, vacuolar isoform (UniProt ID: P32563, chain b), V-type proton ATPase subunit C (UniProt ID: P31412, chain O), and V-type proton ATPase subunit c (UniProt ID: P25515, chains Y, R, U, V, T, W, S, X, Z, and a), a noticeable portion of the subunits of this proteinaceous machine contain regions with unresolved structures [124]. In fact, of 256 residues of chain M (UniProt ID: P32610), residues 1–7 and 218–256 constitute regions with unresolved structures. Also, in other chains of the eukaryotic ATPase, the following residues are structurally uncharacterized: 1 and 117–118 in chain N (UniProt ID: P39111, 118 residues); 1–23 in chains A, C, and E (UniProt ID: P17255, 1,071 residues), 1–28 and 486–517 in chains B, D, and F (UniProt ID: P16140, 517 residues), 1 and 107–114 in chains L, H, and J (UniProt ID: P48836, 114 residues), 1–7 and 225–233 in chains K, G, and I (UniProt ID: P22203, 233 residues), 56–72 in chain P (UniProt ID: P41807, 478 residues), 1–13, 152–175, 221–233, and 363-840 in chain b (UniProt ID: P32563, 840 residues); and 1–10 in chains Y, R, U, V, T, W, S, X, Z, and a (UniProt ID: P25515, 160 residues). Figure 4 illustrates the disorder profiles of the protomers of the eukaryotic ATPase and shows that predicted disorder content in these proteins, ranging from 3.1% to 67.3%. Only one of the 11 protomers shown in Figure 4 is expected to be highly ordered (i.e., shows predicted disorder content (PDC) below 10%), whereas the remaining subunits are either moderately or (10 ≤ PDC < 30%) highly disordered (PDC ≥30%), following the accepted criteria for the classification of disordered proteins [125]. 

### 3.3. Transient Receptor Potential (TRP) Channels 

TRP channels constitute a broad family of diversified non-selective cation channels responding to various chemical and physical stimuli [126,127]. These membrane receptor proteins are present in cells of a large variety of organisms from yeast to humans. They respond to a diverse range of physiological stimuli [36]. They share transmembrane topology and compositional similarities with voltage-gated sodium and potassium (Na_v_ and K_v_) channels, serving as non-selective cationic channels that play important roles in many cellular and physiological processes, such as sensory signal transduction and calcium absorption [128]. TRP channels and, most especially, transient receptor potential cation channel subfamily V member 1 (TRPV1), which is activated by heat and capsaicin (the pungent agent from chili peppers), make up a class of viable molecular targets for the development of pain neutralizing drugs because of their association with pain physiology [129,130]. Therefore, studying the mechanisms by which these channels respond to physiological stimuli is expected to potentiate better understanding of the disorders of major organ systems [128]. 

Like all other mammalian membrane proteins, crystallization of TRP channels remains a challenge. TRP channels respond to a wide range of chemical and physical stimuli and are conformationally heterogeneous. Although NMR and X-ray structures of high resolution have been determined for cytoplasmic domains of TRP channels, a complete atomic model using these approaches still remains an illusion [128,131,132]. With the advent of improved software and hardware components of cryo-EM, previously encountered structural limitations have been bypassed and high-resolution images of pure TRP channels can now be determined at conditions not requiring complete sample homogeneity [128,133]. The structure of the TRPV1 ion channel solubilized in amphipols (which are short amphipathic polymers that can keep membrane proteins water solubility and, therefore, are used to substitute detergents [134]) was resolved to a resolution of ~3.4 Å by single-particle cryo-EM [128]. This high-resolution 3D density map revealed that side chain conformations was first reported in 2013, making TRPV1 the very first membrane protein structure to be determined by single particle cryo-EM at atomic resolution [36]. Later, the TRPV1 ion channel was reconstituted into lipid nano-discs generated with different membrane scaffold proteins (MSPs) [133].

Figure 5A–C shows three projections of the 3D structure of rat TRPV1 resolved by single-particle cryo-EM (PDB ID: 5IRZ, [133]) and illustrate rather intertwined topology of this homo-tetrameric ion channel. Figure 5D represents an intrinsic disorder profile generated for the rat TRPV1 (UniProt ID: O35433) and shows that a significant part of the N-terminal cytoplasmic domain and almost the entire C-terminal cytoplasmic domain are predicted to be highly disordered (PDC = 23.7%), providing an explanation for the failure to determine the crystal structure for the members of the TRP channel family. It is also important to emphasize that the single-particle cryo-EM analysis of rat TRPV1 was conducted using a deletion mutant of this protein with the enhanced biochemical stability consisting of residues 110–764 and excluding the highly divergent region (residues 604–626) [128]. Figure 5D shows that the two regions excluded from structural analysis (residues 1–109 and 604–626) are predicted to be highly disordered. Curiously, this approach (deletion of flexible protein regions) represents a common methodology in sample preparation for X-ray crystallography. Therefore, the presence of high levels of intrinsic disorder limits the application of both the X-ray crystallography and single-particle cryo-EM techniques.

### 3.4. Potassium Channels 

Voltage-activated potassium (K_v_) channels exist in almost all life forms, playing diverse roles. They reside in the cell membrane and have transmembrane regions spanning the full length of the phospholipid bilayer [135] and utilize membrane depolarization as a signal to open and conduct K^+^ ions [136,137]. Because of their ability to respond to electrical signaling, these channels play important roles in both excitable (neurons, muscle cells) and non-excitable (non-muscle and neuronal) cells/tissues. They are responsible for regulating the movement of K^+^ ions in and out of the cells [135]. Membrane depolarization activates voltage-sensing domains of K_v_ channels, leading to the opening of a highly selective central pore domain conducting K^+^ ions [138,139]. Some functional defects originating because of gene mutation events have been associated with the pathophysiology of various diseases, justifying the need for these channels to be carefully studied [140]. 

X-ray crystallography was the dominant tool for structural characterization of K^+^ channels, resulting in a large amount of structures yielding considerable biological information, including structural insights into the mechanism of voltage gating [36]. However, single-particle cryo-EM impacted investigations into the detailed structure of K^+^ channels, especially providing important information on the conformational reorganization of these channels that are associated with the voltage gating, while responding to fluctuations in the membrane potential. Studies have reported the use of various biochemical means, such as high affinity F_ab_s, to stabilize the open conformation of the channel. In addition, liposomal reconstitution was used to modify K^+^ channels to become amenable to cryo-EM [36,141]. An illustrative example of the successful implementation of single-particle cryo-EM in the structural characterization of K_v_ channels is given by the Slo2.2 Na^+^-activated K^+^ channel, which is a member of the Slo family of large conductance K^+^ channels expressed in the brain [142,143]. 

Slo family members contain transmembrane domain (TMD), which is composed of six or seven TM helices and a large C-terminally located cytoplasmic domain (CTD) that includes two RCK (regulator of K^+^ conductance) domains [144]. Despite significant efforts, no structural information was available for any full-length Slo channel until resolving the structure of the homo-tetrameric Slo2.2 Na^+^-activated K^+^ channel (chicken KCNT1, UniProt ID: Q8QFV0) by single-particle cryo-EM [144]. Figure 6A–C represent three projections of this channel (PDB ID: 5U70, [144]). Although the full-length chicken KCNT1 (1201 residues) was used in these experiments, residues 1–68, 119–138, 619–719, 1020–1097, 1136–1137, and 1172–1201 were structurally unassigned [144]. Figure 6D illustrates that chicken KCNT1 (UniProt ID: Q8QFV0, PCD = 24.9%) is predicted to have several long intrinsically disordered regions and a multitude of short disordered and flexible regions. This relatively high disorder content and distribution of disorder predisposition within the amino acid sequence of this channel explains previous failures with structural characterization of the full-length Slo channels by X-ray crystallography. Figure 6D also illustrates that all the structurally unassigned regions of chicken KCNT1 coincide with disordered or flexible regions, indicating that the resolution power of single-particle cryo-EM is limited by high conformational flexibility of target proteins.

## 4. Conclusions

The extensive impact of X-ray crystallography since its inception cannot be overstated, however like many other technologies, it is not without its limitations. In an effort to surmount the challenges associated with the technique, alternative biophysical tools for structure determination were successfully developed based on the foundation laid by a pioneering X-ray crystallography. NMR spectroscopy was developed to handle molecular interactions and analyze dynamic systems. Its reign was short-lived, however, due to the daunting task of resolving copious amounts of data.

The emergence of single particle cryo-EM was a ray of hope for structural determination of biologically relevant macromolecules considered unamenable to existing methods. Consequently, single particle cryo-EM possesses the ability to navigate challenges of older methods in terms of large size and structural heterogeneity. That being said, the idea of cryo-EM completely replacing older methods of structure determination is still a bit far-fetched. An ideal scenario is the complementary use of all three methods discussed in this review.

In fact, as shown in this article, high conformational flexibility and the presence of long disordered regions represent logical limitations of the applicability of single-particle cryo-EM for structural characterization of target proteins. This is because the highly flexible segments may not be observed (or adversely affected) in the cryo-EM. Although this may limit the biological interpretations, it may not completely preclude structural characterization of proteins with flexible segments/regions. Although, this technique represents a promising approach for structural description of large proteinaceous machines with some conformational heterogeneity (e.g., proteins possessing several open and closed conformations), the presence of disordered or highly flexible regions represents a bottleneck for structural characterization of proteins by single-particle Cryo-EM.

## Figures and Tables

**Figure 1 ijms-20-04186-f001:**
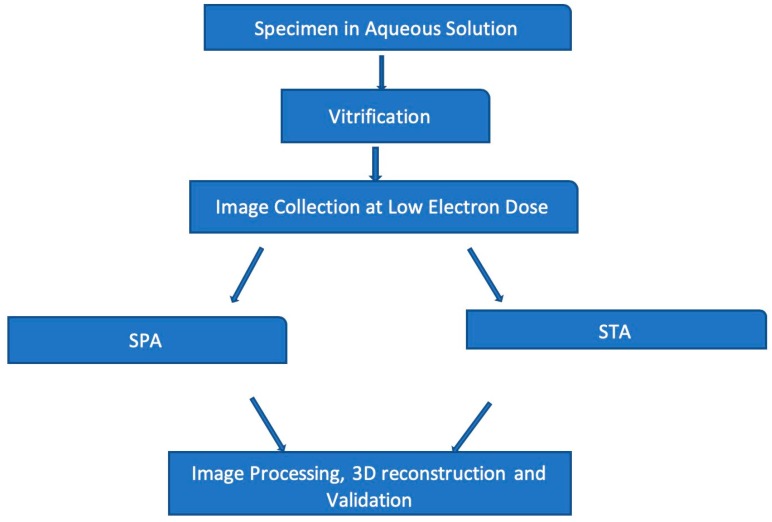
Basic workflow of Cryo-Electron Microscopy (Cryo-EM). Image collected at low dose electron can be analyzed by single-particle or sub-tomogram averaging. However, samples must be vitrified by flash-freezing.

**Figure 2 ijms-20-04186-f002:**
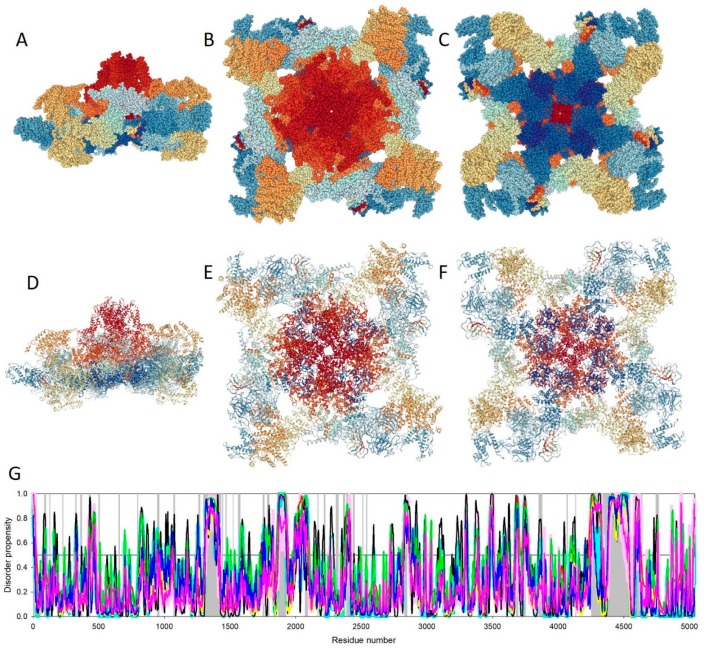
Structural characterization of rabbit RyR1 by single-particle cryo-EM (protein data bank (PDB) ID: 5T15) and a set of commonly used predictors of intrinsic disorder. (**A**). Side view of the channel in a space-fill representation. (**B**). Membrane-side view of the channel in a space-fill representation. (**C**). Cytoplasmic-side view of the channel in a space-fill representation. (**D**). Side view of the channel in a cartoon representation. (**E**). Membrane-side view of the channel in a cartoon representation. (**F**). Cytoplasmic-side view of the channel in a cartoon representation. In these plots, the chains are colored using the rainbow schema of the PDB 3D-viewer (where the N- and C-terminal regions are colored blue and red, respectively). (**G**). Evaluation of the intrinsic disorder propensity of rabbit RyR1 (UniProt ID: P11716) by a set of commonly used disorder predictors. Presented disorder profiles were generated by PONDR-FIT (pink curve), PONDR^®^ VLXT (black curve), PONDR^®^ VSL2 (green curve), and PONDR^®^ VL3 (red curve) [106,107,108,109,110,111], as well as two tools from the IUPred web server for predicting short and long disordered regions (blue and yellow curves, respectively) [112]. The dark cyan dashed line shows the mean disorder propensity calculated by averaging the disorder profiles of the individual predictors. The light pink shadow around the PONDR^®^ FIT shows error distribution, whereas the light cyan shadow around the mean disorder curve reflects the distribution of standard deviations. Light gray bars show positions of structurally uncharacterized regions. In these analyses, the predicted intrinsic disorder scores above 0.5 are considered to correspond to the disordered residues/regions, whereas regions with disorder scores between 0.2 and 0.5 are considered flexible.

**Figure 3 ijms-20-04186-f003:**
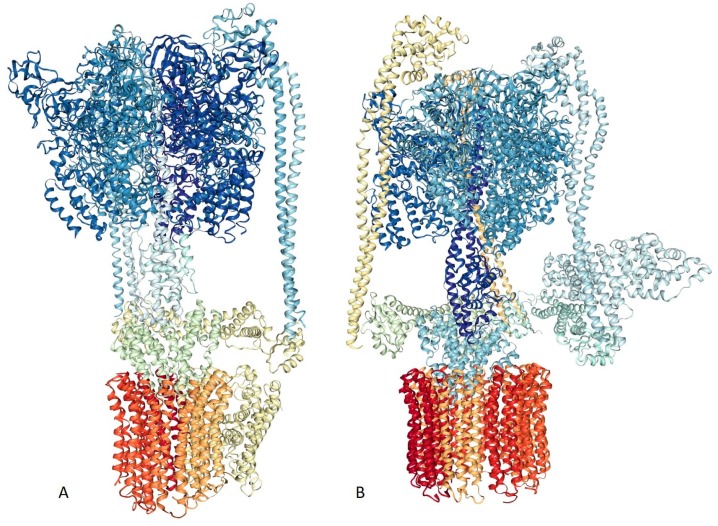
Single-particle cryo-EM-based structural characterization of the eubacterial and eukaryotic vacuolar-type ATPases (V-ATPases) from *Thermus thermophiles* ((**A**). PDB ID: 5GAR, [123]) and *Saccharomyces cerevisiae* ((**B**). PDB ID: 3J9T, [124]), respectively.

**Figure 4 ijms-20-04186-f004:**
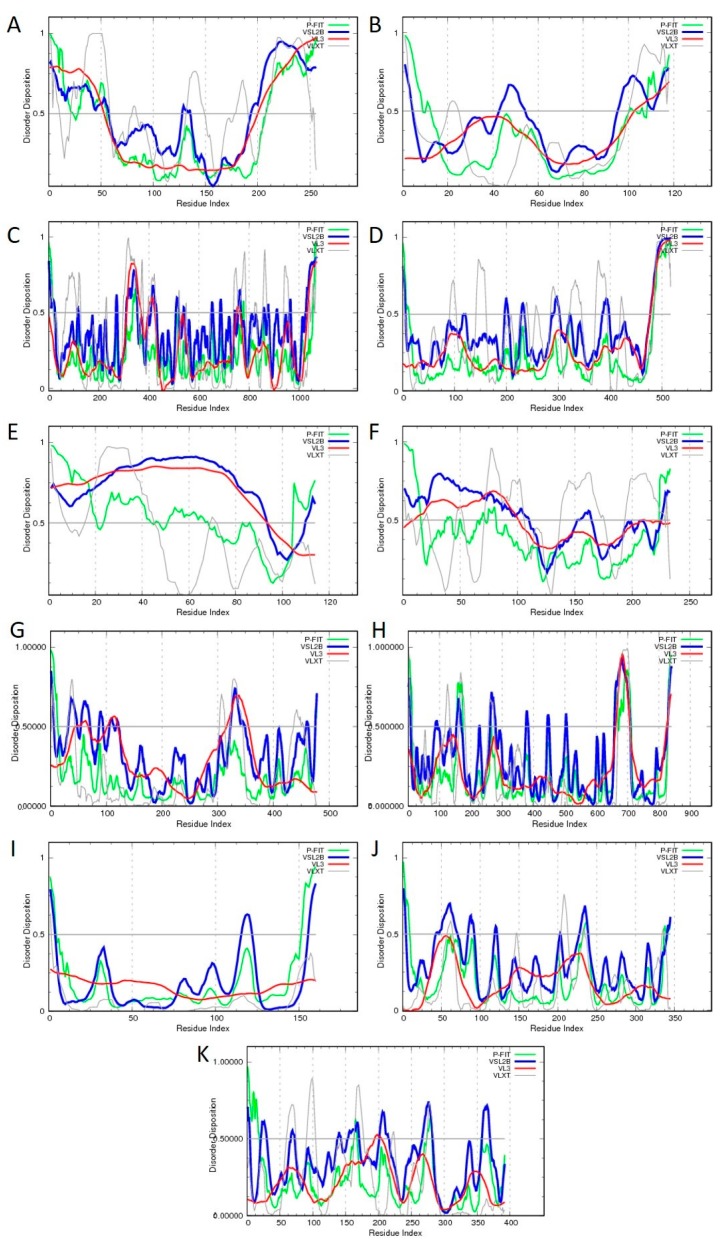
Intrinsic disorder status of the protomers of the V-ATPase from *Saccharomyces cerevisiae* evaluated by PONDR-FIT (green curves), PONDR^®^ VLXT (gray curves), PONDR^®^ VSL2 (blue curves), and PONDR^®^ VL3 (red curves) [106,107,108,109,110,111]. (**A**). V-type proton ATPase subunit D (UniProt ID: P32610, predicted disorder content (PDC) 47.3%); (**B**). V-type proton ATPase subunit F (UniProt ID: P39111, PDC = 22.9%); (**C**). V-type proton ATPase catalytic subunit A (UniProt ID: P17255, PDC = 20.6%); (**D**). V-type proton ATPase subunit B (UniProt ID: P16140, PDC = 17.9%); (**E**). V-type proton ATPase subunit G (UniProt ID: P48836, PDC = 67.3%); (**F**). V-type proton ATPase subunit E (UniProt ID: P22203, PDC = 50.8%); (**G**). V-type proton ATPase subunit H (UniProt ID: P41807, PDC = 17.0%); (**H**). V-type proton ATPase subunit a, vacuolar isoform (UniProt ID: P32563, PDC = 12.5%); (**I**). V-type proton ATPase subunit c (UniProt ID: P25515, PDC = 3.1%); (**J**). V-type proton ATPase subunit d (UniProt ID: P32366, PDC = 12.3%); (**K**). V-type proton ATPase subunit C (UniProt ID: P31412, PDC = 13.2%).

**Figure 5 ijms-20-04186-f005:**
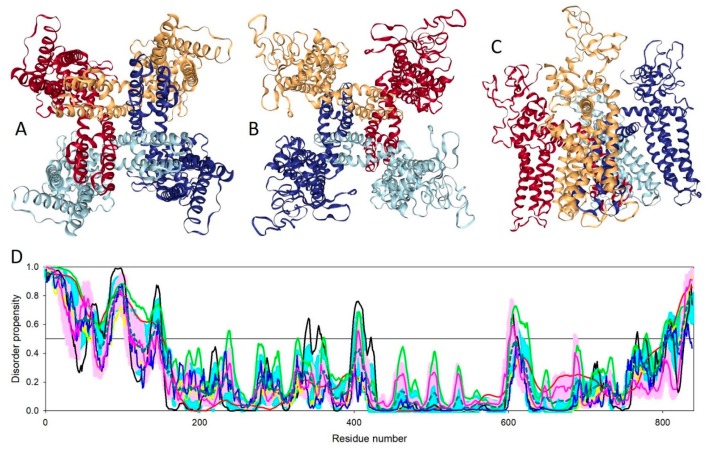
Three projections of the 3D structure of the deletion mutant of rat TRPV1 resolved by single-particle cryo-EM (PDB ID: 5IRZ, [133]): (**A**). Cytoplasm view; (**B**). Membrane view; and (**C**). Side view. (**D**). Evaluation of the intrinsic disorder propensity of the full-length rat TRPV1 (UniProt ID: O35433) by a set of commonly used disorder predictors. Presented disorder profiles were generated by PONDR-FIT (pink curve), PONDR^®^ VLXT (black curve), PONDR^®^ VSL2 (green curve), and PONDR^®^ VL3 (red curve) [106,107,108,109,110,111], as well as two tools from the IUPred web server for predicting short and long disordered regions (blue and yellow curves, respectively) [112]. The dark cyan dashed line shows the mean disorder propensity calculated by averaging the disorder profiles of the individual predictors. The light pink shadow around the PONDR^®^ FIT shows error distribution, whereas the light cyan shadow around the mean disorder curve reflects the distribution of the standard deviations. In these analyses, the predicted intrinsic disorder scores above 0.5 are considered to correspond to the disordered residues/regions, whereas regions with disorder scores between 0.2 and 0.5 are considered flexible.

**Figure 6 ijms-20-04186-f006:**
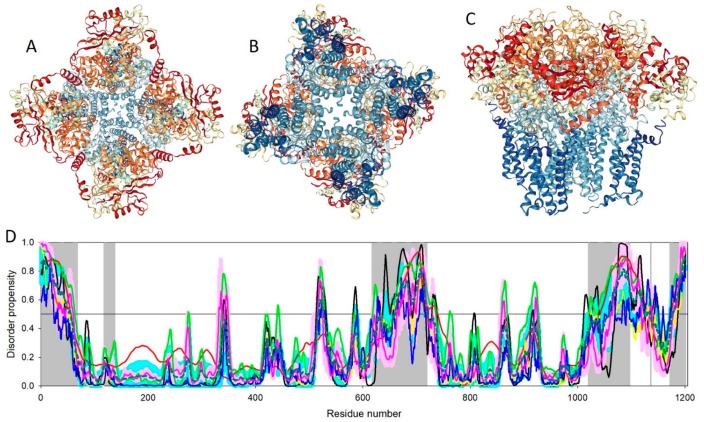
Three projections of the 3D structure of the homo-tetrameric Slo2.2 Na^+^-activated K^+^ channel (chicken KCNT1) resolved by single-particle cryo-EM (PDB ID: 5U70, [144]): (**A**). Cytoplasm view; (**B**). Membrane view; and (**C**). Side view. (**D**). Evaluation of the intrinsic disorder propensity of the chicken KCNT1 (UniProt ID: Q8QFV0) by a set of commonly used disorder predictors. Presented disorder profiles were generated by PONDR-FIT (pink curve), PONDR^®^ VLXT (black curve), PONDR^®^ VSL2 (green curve), and PONDR^®^ VL3 (red curve) [106,107,108,109,110,111], as well as two tools from the IUPred web server for predicting short and long disordered regions (blue and yellow curves, respectively) [112]. The dark cyan dashed line shows the mean disorder propensity calculated by averaging the disorder profiles of the individual predictors. The light pink shadow around the PONDR^®^ FIT shows the error distribution, whereas the light cyan shadow around the mean disorder curve reflects the distribution of the standard deviations. The light gray bars show positions of structurally uncharacterized regions. In these analyses, the predicted intrinsic disorder scores above 0.5 are considered to correspond to the disordered residues/regions, whereas regions with disorder scores between 0.2 and 0.5 are considered flexible.

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
