# Peer review of "Structure Determination by Single-Particle Cryo-Electron Microscopy: Only the Sky (and Intrinsic Disorder) is the Limit"

_ijms, 2019, doi:10.3390/ijms20174186_

Round 1
Reviewer 1 Report
The manuscript is very well written discussing one of the fast-developing techniques in structural biology. I have a few comments/questions:
1- Talking about the limited application of NMR in larger protein: it will be useful to add that there are advances in this field (like methyl TROSY, selective labeling, 13C-reductive methylation, ...)
2- The use of cross-linking for reduction of sample heterogeneity, can the authors comment if this is still widely used way in EM especially with all the drawbacks of such an approach?
3- In lines 441-450: The authors explain the reason for failing to crystallize members of the TRP channel family due to large disorder nature: if the crystallization failed for this particular reason, wouldn't EM face the same challenge as well? Can the authors comment on that?
Author Response
The manuscript is very well written discussing one of the fast-developing techniques in structural biology. I have a few comments/questions:
REPLY: We are thankful to this reviewer for carefull reading of our manuscript, high evaluation of this work and for useful suggestions. We adderess all the critiques and amended manuscript accordingly. We hope that the revised version became more suitable for publication.
1- Talking about the limited application of NMR in larger protein: it will be useful to add that there are advances in this field (like methyl TROSY, selective labeling, 13C-reductive methylation, ...)
REPLY: Thank you for pointing this out. The corresponding clarification is added to the revised manuscript (see page 3 of the revised manuscript).
2- The use of cross-linking for reduction of sample heterogeneity, can the authors comment if this is still widely used way in EM especially with all the drawbacks of such an approach?
REPLY: The corresponding note is added to the revised manuscript (see page 6).
3- In lines 441-450: The authors explain the reason for failing to crystallize members of the TRP channel family due to large disorder nature: if the crystallization failed for this particular reason, wouldn't EM face the same challenge as well? Can the authors comment on that?
REPLY: Thank you for pointing this out. This is a valid point and the presence of high levels of intrinsic disorder will limit application of both techniques, X-ray crystallography and single-particle cryo-EM. As indicated in the manuscript, when structure of rat TRPV1 was solved by single-particle cryo-EM, the authors used a deletion mutant of this protein, where disordered regions were removed. Actually, this approach (deletion of flexible protein regions) represents a common methodology in sample preparation for X-ray crystallography. The corresponding cladification is added to the revised manuscript (see page 15).
Reviewer 2 Report
The authors have given a nice overview of three major structural biology techniques namely X-ray, NMR, and cryo-EM. Since this review is primarily about the cryo-EM, authors have appropriately provided more detailed description of the technique. The review then describes many examples of membrane proteins study by cryo-EM and present a correlation to show that intrinsically disordered or flexible regions are typically not observed in cryo-EM density maps. The analysis is sound, however, it will be clear to re-phrase the conclusion “high conformational flexibility and presence of long disordered regions represent logical limitations of the applicability of single-particle cryo-EM for structural characterization of target proteins.” to indicate that the highly flexible segments may not be observed (or adversely affected) in the cryo-EM that may limit the biological interpretations but may not completely preclude structural characterization of proteins with flexible segments/regions.
Minor comments:
Usage of “sub-nanometer atomic resolutions” should accompany with a relevant resolution value/range like in the page 3, line 97 “sub-nanometer atomic resolutions (< 4 Å )” for clarity.Line 436-438: The sentence “TRPV1 ion channel reconstituted into lipid Nano-discs generated with different membrane scaffold proteins (MSPs) was reconstructed to a resolution of ~3.4 Å by single-particle cryo-EM was first reported in 2013 [113].” Is confusing. In 2013, TRPV1 was solubilized in amphipol – while the nanodisc solubilized TRPV1 was studied in 2016.
Line 444: “a significant part if the N-terminal” should be “a significant part of the N-terminal”
Author Response
The authors have given a nice overview of three major structural biology techniques namely X-ray, NMR, and cryo-EM. Since this review is primarily about the cryo-EM, authors have appropriately provided more detailed description of the technique. The review then describes many examples of membrane proteins study by cryo-EM and present a correlation to show that intrinsically disordered or flexible regions are typically not observed in cryo-EM density maps.
REPLY: We are thankful to this reviewer for carefull reading of our manuscript, high evaluation of this work and for useful suggestions. We adderess all the critiques and amended manuscript accordingly. We hope that the revised version became more suitable for publication.
The analysis is sound, however, it will be clear to re-phrase the conclusion “high conformational flexibility and presence of long disordered regions represent logical limitations of the applicability of single-particle cryo-EM for structural characterization of target proteins.” to indicate that the highly flexible segments may not be observed (or adversely affected) in the cryo-EM that may limit the biological interpretations but may not completely preclude structural characterization of proteins with flexible segments/regions.
REPLY: Thank you for pointing this out. The corresponding clarification was added to the revised manuscript (see page 19).
Minor comments:
Usage of “sub-nanometer atomic resolutions” should accompany with a relevant resolution value/range like in the page 3, line 97 “sub-nanometer atomic resolutions (< 4 Å )” for clarity.
REPLY: This clarification is added (see pages 1 and 12)
Line 436-438: The sentence “TRPV1 ion channel reconstituted into lipid Nano-discs generated with different membrane scaffold proteins (MSPs) was reconstructed to a resolution of ~3.4 Å by single-particle cryo-EM was first reported in 2013 [113].” Is confusing. In 2013, TRPV1 was solubilized in amphipol – while the nanodisc solubilized TRPV1 was studied in 2016.
REPLY: Thank you for pointing this out. The corresponding changes and clarifications are added (see page 15).
Line 444: “a significant part if the N-terminal” should be “a significant part of the N-terminal”
REPLY: Thank you for pointing this out. This misprint was corrected.